# The Traveling Observer Model: Multi-task Learning Through Spatial Variable Embeddings

**Elliot Meyerson**
Cognizant AI Labs
elliot.meyerson@cognizant.com

**Risto Miikkulainen**
UT Austin & Cognizant AI Labs
risto@cs.utexas.edu

## Abstract

This paper frames a general prediction system as an observer traveling around a continuous space, measuring values at some locations, and predicting them at others. The observer is completely agnostic about any particular task being solved; it cares only about measurement locations and their values. This perspective leads to a machine learning framework in which seemingly unrelated tasks can be solved by a single model, by embedding their input and output variables into a shared space. An implementation of the framework is developed in which these variable embeddings are learned jointly with internal model parameters. In experiments, the approach is shown to (1) recover intuitive locations of variables in space and time, (2) exploit regularities across related datasets with completely disjoint input and output spaces, and (3) exploit regularities across seemingly unrelated tasks, outperforming task-specific single-task models and multi-task learning alternatives. The results suggest that even seemingly unrelated tasks may originate from similar underlying processes, a fact that the traveling observer model can use to make better predictions.

## 1 Introduction

Natural organisms benefit from the fact that their sensory inputs and action outputs are all organized in the same space, that is, the physical universe. This consistency makes it easy to apply the same predictive functions across diverse settings. Deep multi-task learning (Deep MTL) has shown a similar ability to adapt knowledge across tasks whose observed variables are embedded in a shared space. Examples include vision, where the input for all tasks (photograph, drawing, or otherwise) is pixels arranged in a 2D plane (Zhang et al., 2014; Misra et al., 2016; Rebuffi et al., 2017); natural language (Collobert & Weston, 2008; Luong et al., 2016; Hashimoto et al., 2017), speech processing (Seltzer & Droppo, 2013; Huang et al., 2015), and genomics (Alipanahi et al., 2015), which exploit the 1D structure of text, waveforms, and nucleotide sequences; and video game-playing (Jaderberg et al., 2017; Teh et al., 2017), where interactions are organized across space and time. Yet, many real-world prediction tasks have no such spatial organization; their input and output variables are simply labeled values, e.g., the height of a tree, the cost of a haircut, or the score on a standardized test. To make matters worse, these sets of variables are often disjoint across a set of tasks. These challenges have led the MTL community to avoid such tasks, despite the fact that general knowledge about how to make good predictions can arise from solving seemingly "unrelated" tasks (Mahmud & Ray, 2008; Mahmud, 2009; Meyerson & Miikkulainen, 2019).

This paper proposes a solution: Learn all variable locations in a shared space, while simultaneously training the prediction model itself (Figure 1). To illustrate this idea, Figure 1a gives an example of four tasks whose variable values are measured at different locations in the same underlying 2D embedding space. The shape of each marker (i.e., $\circ, \square, \triangle, \star$) denotes the task to which that variable belongs; white markers denote input variable, black markers denote output variables, and the background coloring indicates the variable values in the entire embedding space when the current sample is drawn. As a concrete example, the color could indicate the air temperature at each point in a geographical region at a given moment in time, and each marker the location of a temperature sensor (however, note that the embedding space is generally more abstract). Figure 1b-c shows a model that can be applied to any task in this universe, using the $\circ$ task as an example: (b) The function $f$ encodes the value of each observed variable $x_i$ given its 2D location $\mathbf{z}_i \in \mathbb{R}^2$, and these encodings

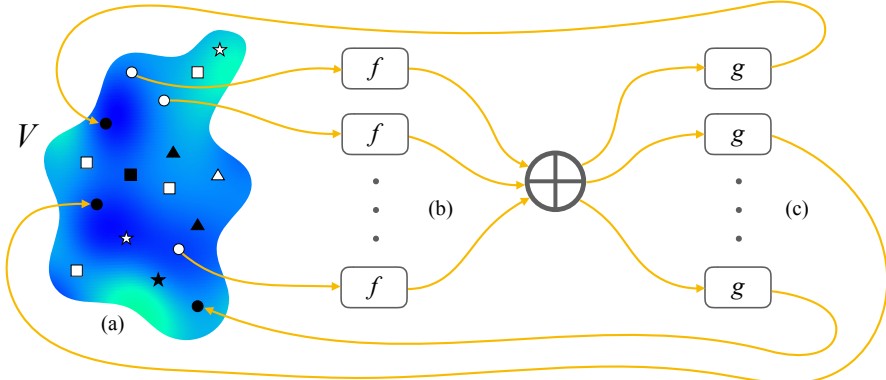

Figure 1: *The Traveling Observer Model.* (a) Tasks with disjoint input and output variable sets are measured in the same underlying 2D universe. The shape of each marker (i.e., $\circ, \square, \triangle, \star$) denotes the task to which that variable belongs; white markers denote input variables, black markers denote output variables, and the background color shows the state of the entire universe when the current sample is drawn. (b) The function $f$ encodes the value of each observed variable $x_i$ given its 2D location $\mathbf{z}_i \in \mathbb{R}^2$, and these encodings are aggregated by elementwise addition $\bigoplus$; (c) The function $g$ decodes the aggregated encoding to a prediction for $y_j$ at its location $\mathbf{z}_j$. In general, the embedded locations $\mathbf{z}$ are not known a priori, but they can be learned alongside $f$ and $g$ by gradient descent.

are aggregated by elementwise addition $\bigoplus$; (c) The function $g$ decodes the aggregated encoding to a prediction for $y_j$ at its location $\mathbf{z}_j$. Such a predictor can be viewed as a *traveling observer model* (TOM): It traverses the space of variables, taking a measurement at the location of each input. Given these observations, the model can make a prediction for the value at the location of an output. In general, the embedded locations $\mathbf{z}$ are not known a priori (i.e., when input and output variables do not have obvious physical locations), but they can be learned alongside $f$ and $g$ by gradient descent.

The input and output spaces of a prediction problem can be standardized so that the measured value of each input and output variable is a scalar. The prediction model can then be completely agnostic about the particular task for which it is making a prediction. By learning *variable embeddings* (VEs), i.e., the $\mathbf{z}$'s, the model can capture variable relationships explicitly and supports joint training of a single architecture across seemingly unrelated tasks with disjoint input and output spaces. TOM thus establishes a new lower bound on the commonalities shared across real-world machine learning problems: They are all drawn from the same space of variables that humans can and do measure.

This paper develops a first implementation of TOM, using an encoder-decoder architecture, with variable embeddings incorporated using FiLM (Perez et al., 2018). In the experiments, the implementation is shown to (1) recover the intuitive locations of variables in space and time, (2) exploit regularities across related datasets with disjoint input and output spaces, and (3) exploit regularities across seemingly unrelated tasks to outperform single-tasks models tuned to each tasks, as well as current Deep MTL alternatives. The results confirm that TOM is a promising framework for representing and exploiting the underlying processes of seemingly unrelated tasks.

## 2 BACKGROUND: MULTI-TASK ENCODER-DECODER DECOMPOSITIONS

This section reviews Deep MTL methods from the perspective of decomposition into encoders and decoders (Table 1). In MTL, there are $T$ tasks $\{(\mathbf{x}_t, \mathbf{y}_t)\}_{t=1}^{T}$ that can, in general, be drawn from different domains and have varying input and output dimensionality. The $t$th task has $n_t$ input variables $[x_{t1}, \dots, x_{tn_t}] = \mathbf{x}_t \in \mathbb{R}^{n_t}$ and $m_t$ output variables $[y_{t1}, \dots, y_{tm_t}] = \mathbf{y}_t \in \mathbb{R}^{m_t}$. Two tasks $(\mathbf{x}_t, \mathbf{y}_t)$ and $(\mathbf{x}_{t'}, \mathbf{y}_{t'})$ are disjoint if their input and output variables are non-overlapping, i.e., $\left(\{x_{ti}\}_{i=1}^{n_t} \cup \{y_{tj}\}_{j=1}^{m_t}\right) \cap \left(\{x_{t'i}\}_{i=1}^{n_{t'}} \cup \{y_{t'j}\}_{j=1}^{m_{t'}}\right) = \varnothing$. The goal is to exploit regularities across task models $\mathbf{x}_t \mapsto \hat{\mathbf{y}}_t$ by jointly training them with overlapping parameters.

The standard *intra-domain* approach is for all task models to share their encoder $f$, and each to have its own task-specific decoder $g_t$ (Table 1a). This setup was used in the original introduction of MTL

| (a) Intra-domain | (b) Task Embeddings | (c) Cross-domain | (d) Variable Embeddings (TOM) |
|---|---|---|---|
| $\hat{\mathbf{y}}_t = g_t(f(\mathbf{x}_t))$ | $\hat{\mathbf{y}}_t = g(f(\mathbf{x}_t, \mathbf{z}_t)))$ | $\hat{\mathbf{y}}_t = g_t(f_t(\mathbf{x}_t))$ | $\hat{y}_j = g\left(\sum_i f(x_i, \mathbf{z}_i), \mathbf{z}_j\right)$ |

Table 1: MTL approaches decomposed into encoders $f_*$ and decoders $g_*$: (a) Standard MTL takes advantage of the shared spatialization of tasks within a domain by sharing a single encoder across all tasks $t$; (b) Task embeddings allow tasks within a domain to share their decoder as well; (c) Applying standard MTL across domains requires task-specific encoders, and finding some other method of sharing parameters across tasks; (d) TOM allows a single encoder and decoder to be used even in the cross-domain setting, by embedding all input and output variables into a shared space.

(Caruana, 1998), has been broadly explored in the linear regime (Argyriou et al., 2008; Kang et al., 2011; Kumar & Daumé, 2012), and is the most common approach in Deep MTL (Huang et al., 2013; Zhang et al., 2014; Dong et al., 2015; Liu et al., 2015; Ranjan et al., 2016; Jaderberg et al., 2017). The main limitation of this approach is that it is limited to sets of tasks that are all drawn from the same domain. It also has the risk of the separate decoders doing so much of the learning that there is not much left to be shared, which is why the decoders are usually single affine layers.

To address the issue of limited sharing, the *task embeddings* approach trains a single encoder $f$ and single decoder $g$, with all task-specific parameters learned in embedding vectors $\mathbf{z}_t$ that semantically characterize each task, and which are fed into the model as additional input (Yang & Hospedales, 2014; Bilen & Vedaldi, 2017; Zintgraf et al., 2019) (Table 1b). Such methods require that all tasks have the same input and output space, but are flexible in how the embeddings can be used to adapt the model to each task. As a result, they can learn tighter connections between tasks than separate decoders, and these relationships can be analyzed by looking at the learned embeddings.

To exploit regularities across tasks from diverse and disjoint domains, *cross-domain* methods have been introduced. Existing methods address the challenge of disjoint output and input spaces by using separate decoders *and* encoders for each domain (Table 1c), and thus they require some other method of sharing model parameters across tasks, such as sharing some of their layers (Kaiser et al., 2017; Meyerson & Miikkulainen, 2018) or drawing their parameters from a shared pool (Meyerson & Miikkulainen, 2019). For many datasets, the separate encoder and decoder absorbs too much functionality to share optimally, and their complexity makes it difficult to analyze the relationships between tasks. Earlier work prior to deep learning showed that, from an algorithmic learning theory perspective, sharing knowledge across tasks should always be useful (Mahmud & Ray, 2008; Mahmud, 2009), but the accompanying experiments were limited to learning biases in a decision tree generation process, i.e., the learned models themselves were not shared across tasks.

TOM extends the notion of task embeddings to *variable embeddings* in order to apply the idea in the cross-domain setting (Table 1d). The method is described in the next section.

## 3  THE TRAVELING OBSERVER MODEL

Consider the set of all scalar random variables that could possibly be measured $\{v_1, v_2, ...\} = V$. Each $v_i \in V$ could be an input or output variable for some prediction task. To characterize each $v_i$ semantically, associate with it a vector $\mathbf{z}_i \in \mathbb{R}^C$ that encodes the meaning of $v_i$, e.g., "height of left ear of human adult in inches", "answer to survey question 9 on a scale of 1 to 5", "severity of heart disease", "brightness of top-left pixel of photograph", etc. This vector $\mathbf{z}_i$ is called the *variable embedding* (VE) of $v_i$. Variable embeddings could be handcoded, e.g., based on some featurization of the space of variables, but such a handcoding is usually unavailable, and would likely miss some of the underlying semantic regularities across variables. An alternative approach is to learn variable embeddings based on their utility in solving prediction problems of interest.

A prediction task $(\mathbf{x}, \mathbf{y}) = ([x_1, \ldots, x_n], [y_1, \ldots, y_m])$ is defined by its set of observed variables $\{x_i\}_{i=1}^n \subseteq V$ and its set of target variables $\{y_j\}_{j=1}^m \subseteq V$ whose values are unknown. The goal is to find a prediction function $\Omega$ that can be applied across any prediction task of interest, so that it can learn to exploit regularities across such problems. Let $\mathbf{z}_i$ and $\mathbf{z}_j$ be the variable embeddings corresponding to $x_i$ and $y_j$, respectively. Then, this universal prediction model is of the form

$$\mathbb{E}[y_j \mid \mathbf{x}] = \Omega(\mathbf{x}, \{\mathbf{z}_i\}_{i=1}^n, \mathbf{z}_j). \tag{1}$$

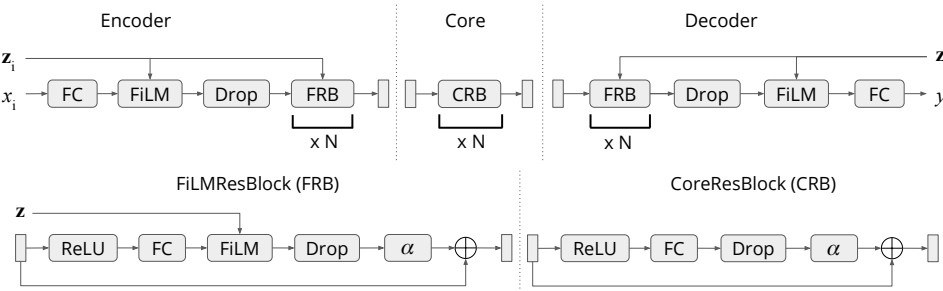

Figure 2: Diagram of the TOM implementation used in the experiments. *Encoder*, *Core*, and *Decoder* correspond to $f$, $g_1$, and $g_2$ in Eq. 4, resp. The Encoder and Decoder are conditioned on input and output VEs $\mathbf{z}$ via FiLM layers. A CRB is simply an FRB without conditioning. Dropout and trainable scalars $\alpha$ implement SkipInit as a substitute for BatchNorm. This residual structure allows the architecture to learn tasks of varying complexity in a flexible manner.

Importantly, for any two tasks $(\mathbf{x}_t, \mathbf{y}_t), (\mathbf{x}_{t'}, \mathbf{y}_{t'})$, their prediction functions (Eq. 1) differ only in their $\mathbf{z}$'s, which enforces the constraint that functionality is otherwise completely shared across the models. One can view $\Omega$ as a *traveling observer*, who visits several locations in the $C$-dimensional variable space, takes measurements at those locations, and uses this information to make predictions of values at other locations.

To make $\Omega$ concrete, it must be a function that can be applied to any number of variables, can fit any set of prediction problems, and is invariant to variable ordering, since we cannot in general assume that a meaningful order exists. These requirements lead to the following decomposition:

$$\mathbb{E}[y_j \mid \mathbf{x}] = \Omega(\mathbf{x}, \{\mathbf{z}_i\}_{i=1}^n, \mathbf{z}_j) = g\Big( \sum_{i=1}^n f(x_i, \mathbf{z}_i), \ \mathbf{z}_j \Big), \tag{2}$$

where $f$ and $g$ are functions called the *encoder* and *decoder*, with trainable parameters $\theta_f$ and $\theta_g$, respectively. The variable embeddings $\mathbf{z}$ tell $f$ and $g$ which variables they are observing, and these $\mathbf{z}$ can be learned by gradient descent alongside $\theta_f$ and $\theta_g$. A depiction of the model is shown in Figure 1. For some integer $M$, $f : \mathbb{R}^{C+1} \to \mathbb{R}^M$ and $g : \mathbb{R}^{M+C} \to \mathbb{R}$. In principle, $f$ and $g$ could be any sufficiently expressive functions of this form. A natural choice is to implement them as neural networks. They are called the encoder and decoder because they map variables to and from a latent space of size $M$. This model can then be trained end-to-end with gradient descent. A batch for gradient descent is constructed by sampling a prediction problem, e.g., a task, from the distribution of problems of interest, and then sampling a batch of data from the data set for that problem. Notice that, in addition to supervised training, in this framework it is natural to autoencode, i.e., predict input variables, and subsample inputs to simulate multiple tasks drawn from the same universe.

The question remains: How can $f$ and $g$ be designed so that they can sufficiently capture a broad range of prediction behavior, and be effectively conditioned by variable embeddings? The next section introduces an experimental architecture that satisfies these requirements.

## 4   INSTANTIATION

The experiments in this paper implement TOM using a generic architecture built from standard components (Figure 2). The encoder and decoder are conditioned on VEs via FiLM layers (Perez et al., 2018), which provide a flexible yet inexpensive way to adapt functionality to each variable, and have been previously used to incorporate task embeddings (Vuorio et al., 2019; Zintgraf et al., 2019). For simplicity, the FiLM layers are based on affine transformations of VEs. Specifically, the $\ell$th FiLM layer $F_\ell$ is parameterized by affine layers $W_\ell^*$ and $W_\ell^+$, and, given a variable embedding $\mathbf{z}$, the hidden state $\mathbf{h}$ is modulated by

$$F_\ell(\mathbf{h}) = W_\ell^*(\mathbf{z}) \odot \mathbf{h} + W_\ell^+(\mathbf{z}), \tag{3}$$

where $\odot$ is the Hadamard product. A FiLM layer is located alongside each fully-connected layer in the encoder and decoder, both of which consist primarily of residual blocks. To avoid deleterious behavior of batch norm across diverse tasks and small datasets/batches, the recently proposed SkipInit

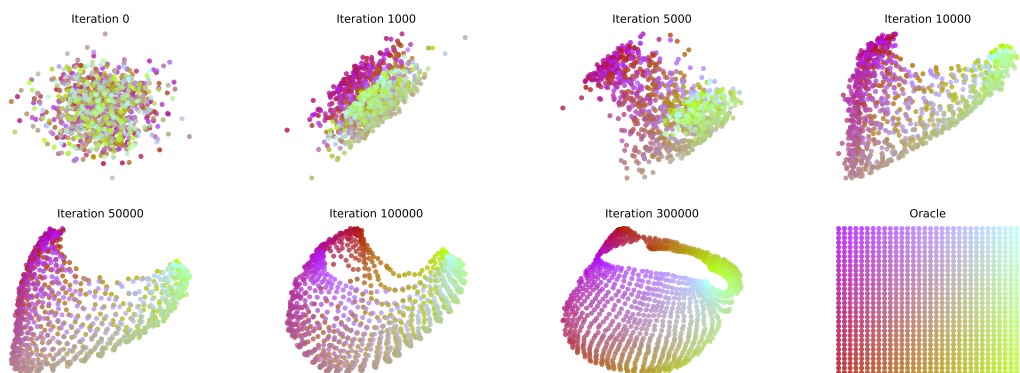

Figure 3: Variable embeddings learned for CIFAR unfold over iterations until they resemble Oracle expectations *(best viewed in color)*. The VE for each variable, i.e., pixel, is colored uniquely. TOM peels the border of the CIFAR images (the upper loop of VEs at iteration 300K) away from their center (the lower grid). This makes sense, since CIFAR images all feature a central object, which semantically splits the image into foreground (the object itself) and background (the remaining ring of pixels around the object). See `https://youtu.be/R_z-2SR2KpY` for videos of VEs being learned.

(De & Smith, 2020) is used as a replacement to stabilize training. SkipInit adds a trainable scalar $\alpha$ initialized to 0 at the end of each residual block, and uses dropout for regularization. Finally, for computational efficiency, the decoder is redecomposed into the Core, or $g_1$, which is independent of output variable, and the Decoder proper, or $g_2$, which is conditioned on the output variable. That way, generic transformations of the summed Encoder output can be learned by the Core and run in a single forward and backward pass each iteration. With this decomposition, Eq. 2 is rewritten as

$$\mathbb{E}[y_j \mid \mathbf{x}] = g_2\Big(g_1\Big(\sum_{i=1}^{n} f(x_i, \mathbf{z}_i)\Big), \, \mathbf{z}_j\Big). \tag{4}$$

The complete architecture is depicted in Figure 2. In the following sections, all models are implemented in pytorch (Paske et al., 2017), use Adam for optimization (Kingma & Ba, 2014), and have hidden layer size of 128 for all layers. Variable embeddings for TOM are initialized from $\mathcal{N}(0, 10^{-3})$. See Appendix C for additional details of this implementation.

## 5 EXPERIMENTS

This section presents a suite of experiments that evaluate the behavior of the implementation introduced in Section 4. See Appendix for additional experimental details.

### 5.1 VALIDATING LEARNED VARIABLE EMBEDDINGS: DISCOVERING SPACE AND TIME

The experiments in this section test TOM's ability to learn variable embeddings that reflect our a priori intuition about the domain, in particular, the organization of space and time.

*CIFAR.* The first experiment is based on the CIFAR dataset (Krizhevsky, 2009). The pixels of the $32 \times 32$ images are converted to grayscale values in [0, 1], yielding 1024 variables. The goal is to predict all variable values, given only a subset of them as input. The model is trained to minimize the binary cross-entropy of each output, and it uses 2D VEs. The a priori, or *Oracle*, expectation is that the VEs form a $32 \times 32$ grid corresponding to how pixels are spatially laid out in an image.

*Daily Temperature.* The second experiment is based on the Melbourne minimum daily temperature dataset (Brownlee, 2016), a subset of a larger database for tracking climate change (Della-Marta et al., 2004). As above, the goal is to predict the daily temperature of the previous 10 days, given only some subset of them, by minimizing the MSE of each variable. The a priori, *Oracle*, expectation is that the VEs are laid out linearly in a single temporal dimension. The goal is to see whether TOM will also learn VEs (in a 2D space) that follow a clear 1D manifold that can be interpreted as time.

| Variable Embeddings | Zero | Random | Learned | Oracle |
|---|---|---|---|---|
| CIFAR (Binary Cross-entropy) | 0.662 ±0.0000 | 0.660 ±0.0007 | **0.591** ±0.0002 | 0.590 ±0.0001 |
| Daily Temperature (RMSE) | 4.29 ±0.002 | 4.27 ±0.011 | **3.32** ±0.011 | 3.37 ±0.005 |

Table 2: *Quantitative results for space and time prediction.* This table compares test errors (± std. err.) of learned VEs to fixed-VE alternatives in TOM. The results show that learned VEs outperform Zero and Random VEs, reaching performance on par with the Oracle. That is, TOM not only learns meaningful VEs (Figures 3 and 4), but also uses these VEs to achieve superior peformance.

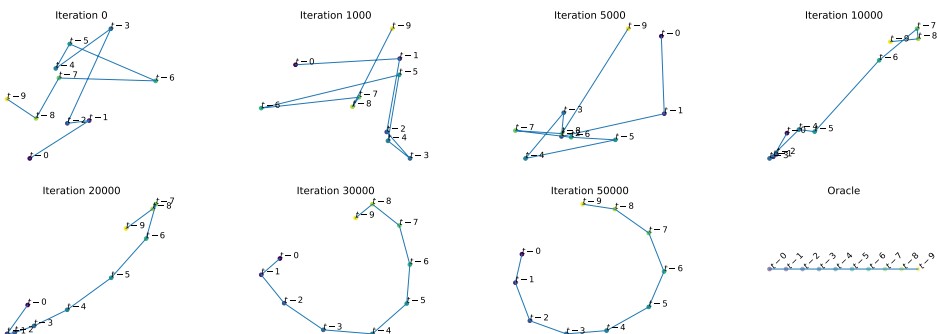

Figure 4: Variable embeddings learned for daily temperature variables untangle over iterations and converge on a 1D manifold ordered by time, as one would expect (neighboring time-steps are connected to illustrate the order). TOM has embedded this 1D structure as a ring in 2D, which is well-suited to the nonlinear encoder and decoder, since it mirrors an isotropic Gaussian distribution.

For both experiments, a subset of the input variables is randomly sampled at each training iteration, which simulates drawing tasks from a limited universe. The resulting learning process for the VEs is illustrated in Figures 3 and 4. The VEs for CIFAR pull apart and unfold, until they reflect the oracle embeddings (Figure 3). The remaining difference is that TOM peels the border of the CIFAR images (the upper loop of VEs at iteration 300K) away from their center (the lower grid). This makes sense, since CIFAR images all feature a central object, which semantically splits the image into foreground (the object itself) and background (the remaining ring of pixels around the object). Similarly, the VEs for daily temperature pull apart until they form a perfect 1D manifold representing the time dimension (Figure 4). The main difference is that TOM has embedded this 1D structure as a ring in 2D, which is well-suited to the nonlinear encoder and decoder, since it mirrors an isotropic Gaussian distribution. Note that unlike visualization methods like SOM (Kohonen, 1990), PCA (Pearson, 1901), or t-SNE (van der Maaten & Hinton, 2008), TOM learns locations for each *variable* not each *sample*. Furthermore, TOM has no explicit motivation to visualize; learned VEs are simply the locations found to be useful by using gradient descent when solving the prediction problem.

To get an idea of how learning VEs affects prediction performance, comparisons were run with three cases of fixed VEs: (1) all VEs set to *zero*, to address the question of whether differentiating variables with VEs is needed at all in the model; (2) *random* VEs, to address the question of whether simply having any unique label for variables is sufficient; and (3) *oracle* VEs, which reflect the human a priori expectation of how the variables should be arranged. The results show that the learned embeddings outperform zero and random embeddings, achieving performance on par with the Oracle (Table 2). The conclusion is that learned VEs in TOM are not only meaningful, but can help make superior predictions, without a priori knowledge of variable meaning. The next section shows how such VEs can be used to exploit regularities across tasks in an MTL setting.

## 5.2 EXPLOITING REGULARITIES ACROSS DISJOINT TASKS

This section considers two synthetic multi-task problems that contain underlying regularities across tasks. These regularities are not known to the model a priori; it can only exploit them via its VEs. The first problem evaluates TOM in a regression setting where input and output variables are drawn from the same continuous space; the second problem evaluates TOM in a classification setting. For classification tasks, each class defines a distinct output variable.

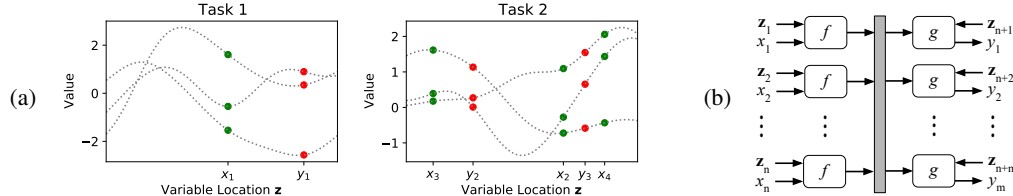

Figure 5: (a) Tasks with disjoint input and output variable sets, whose variables are nonetheless measured in the same underlying space (dotted lines are samples). These tasks are drawn from the Transposed Gaussian Process problem in Section 5.2; (b) TOM can be applied to any task in this space: It predicts values at output locations, given values at input locations.

| Method | Transposed GP (MSE) | Concentric Hyperspheres (Accuracy) |
|---|---|---|
| DR-STL | 0.373 $\pm$0.030 | 42.56 $\pm$1.69 |
| TOM-STL | 0.552 $\pm$0.027 | 64.52 $\pm$1.83 |
| DR-MTL | 0.397 $\pm$0.032 | 54.42 $\pm$1.92 |
| SLO | 0.568 $\pm$0.028 | 53.26 $\pm$1.91 |
| TOM | **0.346** $\pm$0.031 | **92.90** $\pm$1.49 |
| Oracle | 0.342 $\pm$0.026 | 99.96 $\pm$0.02 |

Table 3: *Quantitative Results in synthetic disjoint MTL scenarios.* TOM learns variable embeddings that enable it to outperform alternative approaches, and achieve performance on par with the Oracle.

*Transposed Gaussian Process.* In the first problem, the universe is defined by a Gaussian process (GP). The GP is 1D, is zero-mean, and has an RBF kernel with length-scale 1. One task is generated for each (# inputs, # outputs) pair in $\{1, \ldots, 10\} \times \{1, \ldots, 10\}$, for a total of 100 tasks. The "true" location of each variable lies in the single dimension of the GP, and is sampled uniformly from $[0, 5]$. Samples for the task are generated by sampling from the GP, and measuring the value at each variable location. The dataset for each task contains 10 training samples, 10 validation samples, and 100 test samples. Samples are generated independently for each task. The goal is to minimize MSE of the outputs. Figure 5 gives two examples of tasks drawn from this universe. This testbed is ideal for TOM, because, by the definition of the GP, it explicitly captures the idea that variables whose VEs are nearby are closely related, and every variable has some effect on all others.

*Concentric Hyperspheres.* In the second problem, each task is defined by a set of concentric hyperspheres. Many areas of human knowledge have been organized abstractly as such hyperspheres, e.g., planets around a star, electrons around an atom, social relationships around an individual, or suburbs around Washington D.C.; the idea is that a model that discovers this common organization could then share general knowledge across such areas more effectively. To test this hypothesis, one task is generated for each (# features $n$, # classes $m$) pair in $\{1, \ldots, 10\} \times \{2, \ldots, 10\}$, for a total of 90 tasks. For each task, its origin $\mathbf{o}_t$ is drawn from $\mathcal{N}(\mathbf{0}, I_n)$. Then, for each class $c \in \{1, \ldots, m\}$, samples are drawn from $\mathbb{R}^n$ uniformly at distance $c$ from $\mathbf{o}_t$, i.e., each class is defined by a (hyper) annulus. The dataset for each task contains five training samples, five validation samples, and 100 test samples per class. The model has no a priori knowledge that the classes are structured in annuli, or which annulus corresponds to which class, but it is possible to achieve high accuracy by making analogies of annuli across tasks, i.e., discovering the underlying structure of this universe.

In these experiments, TOM is compared to five alternative methods: (1) *TOM-STL*, i.e. TOM trained on each task independently; (2) *DR-MTL* (Deep Residual MTL), the standard cross-domain (Table 1c) version of TOM, where instead of FiLM layers, each task has its own linear encoder and decoder layers, and all residual blocks are CoreResBlocks; (3) *DR-STL*, which is like DR-MTL except it is trained on each task independently; (4) *SLO* (Soft Layer Ordering; Meyerson & Miikkulainen, 2018), which uses a separate encoder and decoder for each task, and which is (as far as we know) the only prior Deep MTL approach that has been applied across disjoint tabular datasets; and (5) *Oracle*, i.e. TOM with VEs fixed to intuitively correct values. The Oracle is included to give an upper bound on how well the TOM architecture in Section 4 could possibly perform. The oracle VE for each Transposed GP task variable is the location where it is measured in the GP; for Concentric Hyperspheres, the oracle VE for each class $c$ is $c/10$, and for the $i$th feature is $o_i^t$.

TOM outperforms the competing methods and achieves performance on par with the Oracle (Table 3). Note that the improvement of TOM over TOM-STL is much greater than that of DR-MTL over DR-STL, indicating that TOM is particularly well-suited to exploiting structure across disjoint

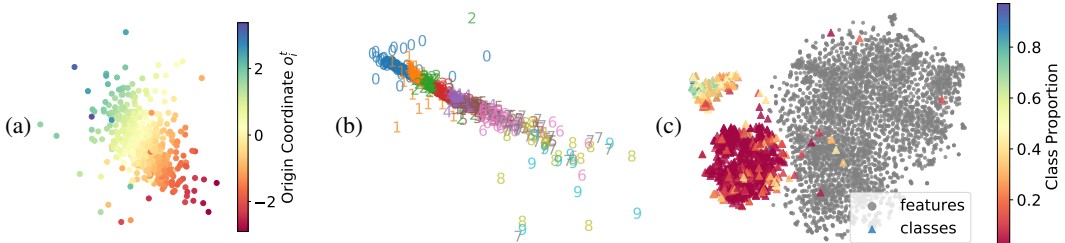

Figure 6: *Learned VEs capture underlying structure across tasks.* (a) VEs of features for concentric hyperspheres encode the origin location, and (b) for classes encode the index of their annuli (less precisely for the more distant annuli, since they occur in fewer tasks); (c) VEs for UCI-121 (shown in 2D via t-SNE) neatly carve the space into features, common classes, and uncommon classes.

data sets (learned VEs are shown in Figure 6a-b). Now that this suitability has been confirmed, the next section evaluates TOM across a suite of disjoint, and seemingly unrelated, real-world problems.

## 5.3   Multi-task learning across seemingly unrelated real-world datasets

This section evaluates TOM in the setting for which it was designed: learning a single shared model across seemingly unrelated real-world datasets. The set of tasks used is UCI-121 (Lichman, 2013; Fernández-Delgado et al., 2014), a set of 121 classification tasks that has been previously used to evaluate the overall performance of a variety of deep NN methods (Klambauer et al., 2017). The tasks come from diverse areas such as medicine, geology, engineering, botany, sociology, politics, and game-playing. Prior work has tuned each model to each task individually in the single-task regime; no prior work has undertaken learning of all 121 tasks in a single joint model. The datasets are highly diverse. Each simply defines a classification task that a machine learning practitioner was interested in solving. The number of features for a task range from 3 to 262, the number of classes from 2 to 100, and the number of samples from 10 to 130,064. To avoid underfitting to the larger tasks, $C = 128$, and after joint training all model parameters ($\theta_f$, $\theta_{g_1}$, $\theta_{g_2}$, and $\mathbf{z}$'s) are finetuned on each task with at least 5K samples. Note that it is not expected that training any two tasks jointly will improve performance in both tasks, but that training all 121 tasks jointly will improve performance overall, as the model learns general knowledge about how to make good predictions.

Results across a suite of metrics are shown in Table 4. *Mean Accuracy* is the test accuracy averaged across all tasks. *Normalized Accuracy* scales the accuracy within each task before averaging across tasks, with 0 and 100 corresponding to the lowest and highest accuracies. *Mean Rank* averages the method's rank across tasks, where the best method gets a rank of 0. *Best %* is the percentage of tasks for which the method achieves the top accuracy (with possible ties). *Win %* is the percentage of tasks for which the method achieves accuracy strictly greater than all other methods. TOM outperforms the alternative approaches across all metrics, showing its ability to learn many seemingly unrelated tasks successfully in a single model (see Figure 6c for a high-level visualization of learned VEs). In other words, TOM can both learn meaningful VEs and use them to improve prediction performance.

## 6   Discussion and Future Work

Sections 2 and 3 developed the foundations for the TOM approach; Sections 4 and 5 illustrated its capabilities, demonstrating its value as a general multitask learning system. This section discusses four key areas of future work for increasing the understanding and applicability of the approach.

First, there is an opportunity to develop a theoretical framework for understanding when TOM will work best. It is straightforward to extend universal approximation results from approximation of single functions (Cybenko, 1989; Lu et al., 2017; Kidger & Lyons, 2020) to approximation of a set of functions each with any input and output dimensionality via Eq. 2. It is also straightforward to extend convergence bounds for certain model classes, such as PAC bounds (Bartlett & Mendelson, 2002; Neyshabur et al., 2018), to TOM architectures implemented with these classes, if the "true" variable embeddings are fixed a priori, so they can simply be treated as features. However, a more

| | Method | Win % | Best % | Mean Rank | Norm. Acc. | Mean Acc. |
|---|---|---|---|---|---|---|
| | ResNet | 3.31 ±1.63 | 12.40 ±3.03 | 3.89 ±0.19 | 50.07 ±3.15 | 79.24 ±1.59 |
| | MS | 4.96 ±1.98 | 14.88 ±3.28 | 3.35 ±0.19 | 60.11 ±3.00 | 80.11 ±1.48 |
| | BN | 5.79 ±2.13 | 13.22 ±3.11 | 4.20 ±0.20 | 42.15 ±3.24 | 77.01 ±1.83 |
| (a) | WN | 7.44 ±2.40 | 10.74 ±2.84 | 4.05 ±0.20 | 45.87 ±3.11 | 77.43 ±1.74 |
| | HW | 8.26 ±2.51 | 15.70 ±3.35 | 3.61 ±0.21 | 53.00 ±3.20 | 78.68 ±1.61 |
| | LN | 9.92 ±2.73 | 16.53 ±3.40 | 3.45 ±0.20 | 56.73 ±3.03 | 79.85 ±1.53 |
| | SNN | 13.22 ±3.09 | 21.49 ±3.78 | 2.78 ±0.19 | 65.29 ±2.84 | 81.39 ±1.35 |
| | TOM | **28.93** ±4.14 | **34.71** ±4.36 | **2.60** ±0.22 | **70.72** ±3.02 | **81.53** ±1.44 |
| | DR-STL | 10.74 ±2.82 | 19.01 ±3.60 | 2.31 ±0.12 | 54.72 ±3.51 | 76.48 ±1.68 |
| | TOM-STL | 7.44 ±2.40 | 16.53 ±3.40 | 2.72 ±0.13 | 35.21 ±3.72 | 68.18 ±2.26 |
| (b) | DR-MTL | 9.09 ±2.62 | 28.10 ±4.12 | 2.02 ±0.12 | 56.47 ±3.68 | 78.40 ±1.47 |
| | SLO | 16.53 ±3.39 | 30.06 ±4.22 | 1.62 ±0.10 | 73.88 ±2.93 | 80.31 ±1.38 |
| | TOM | **32.23** ±4.27 | **47.10** ±4.58 | **1.34** ±0.13 | **76.70** ±3.08 | **81.53** ±1.44 |

Table 4: *UCI-121 Results*. (a) Comparisons to external results of deep STL models tuned to each task (see "Experiments" in Klambauer et al. (2017) for more details); (b) Comparisons across methods evaluated in this paper. Metrics are aggregated over all 121 tasks (± std. err.). TOM achieves high performance across seemingly unrelated tasks, outperforming the comparisons across all metrics.

intriguing direction involves understanding how the true locations of variables affects TOM's ability to learn and exploit them, i.e., what are desirable theoretical properties of the space of variables?

Second, in this paper, TOM was evaluated only in the case when the data for all tasks is always available, and the model is trained simultaneously across all tasks. However, it would also be natural to apply TOM in a meta-learning regime (Finn et al., 2017; Zintgraf et al., 2019), in which the model is trained explicitly to generalize to future tasks, and to lifelong learning (Thrun & Pratt, 2012; Brunskill & Li, 2014; Abel et al., 2018), where the model must learn new tasks as they appear over time. Simply freezing the learned parameters of TOM results in a parametric class of ML models with $C$ parameters per variable that can be applied to new tasks. However, in practice, it should be possible to improve upon this approach by taking advantage of more sophisticated fine-tuning and parameter adaptation. For example, in low-data settings, methods can be adapted from meta-learning approaches that modulate model weights in a single forward pass instead of performing supervised backpropagation (Garnelo et al., 2018; Vuorio et al., 2019). Interestingly, although they are designed to address issues quite different from those motivating TOM, the architectures of such approaches have a functional decomposition that is similar to that of TOM at a high level (see e.g. Conditional Neural Processes, or CNPs; Garnelo et al., 2018). In essence, replacing the VEs in Eq. 2 with input samples and the variables with output samples yields a function that generates a prediction model given a dataset. This analogy suggests that it should be possible to extend the benefits of CNPs to TOM, including rich uncertainty information.

Third, to make the foundational case for TOM, this paper focused on the setting where VEs are a priori unknown, but when such knowledge is available, it could be useful to integrate with learned VEs. Such an approach could eliminate the cost of relearning VEs, and suggest how to take advantage of spatially-customized architectures. E.g., convolution or attention layers could be used instead of dense layers as architectural primitives, as in vision and language tasks. Such specialization could be instrumental in making TOM more broadly applicable and more powerful in practice.

Finally, one interpretation of Fig. 6c is that the learned VEs of classes encode a task-agnostic concept of "normal" vs. "abnormal" system states. TOM could be used to analyze the emergence of such general concepts and as an *analogy engine*: to describe states of one task in the language of another.

## 7 CONCLUSION

This paper introduced the traveling observer model (TOM), which enables a single model to be trained across diverse tasks by embedding all task variables into a shared space. The framework was shown to discover intuitive notions of space and time and use them to learn variable embeddings that exploit knowledge across tasks, outperforming single- and multi-task alternatives. Thus, learning a single function that cares only about variable locations and their values is a promising approach to integrating knowledge across data sets that have no a priori connection. The TOM approach thus extends the benefits of multi-task learning to broader sets of tasks.

ACKNOWLEDGEMENTS

Thank you to Babak Hodjat and others in the Evolutionary AI research group for helpful discussions and technical feedback. Thank you also to the reviewers, particularly for their suggestions for improving the organizational structure and clarity of the paper.

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

## A  ADDITIONAL EXPERIMENT ON THE EMBEDDING SIZE $C$

In the experiments in Section 5.1 and 5.2, the VE dimensionality $C$ for TOM was set to 2 in order to most clearly visualize the VEs that were learned. In the experiment in Section 5.3, $C$ was increased in order to accommodate the scale-up to a large number of highly diverse real world tasks. In that experiment $C$ was set to 128 in order to match the number of task-specific parameters of the other Deep MTL methods compared in Table 4.

To evaluate the sensitivity of TOM to the setting of $C$, additional experiments were run for TOM on UCI-121 with $C = 64$ and $C = 256$. The results are shown in Table 5. Metrics for all settings of $C$ are computed w.r.t. the external comparison methods, i.e., those in Table 4a. TOM with $C = 64$ produces performance comparable to $C = 128$, suggesting that optimizing $C$ could be a useful lever for balancing performance and VE interpretability.

| $C$ | Win % | Best % | Mean Rank | Norm. Acc. | Mean Acc. |
|-----|-------|--------|-----------|------------|-----------|
| 256 | 21.49 | 24.79 | 2.96 | 65.24 | 80.68 |
| 128 | **28.93** | **34.71** | 2.60 | 70.72 | 81.53 |
| 64 | 25.62 | 32.23 | **2.50** | **71.28** | **81.97** |

Table 5: Results for TOM on UCI-121 with varying VE dimensionality $C$.

## B  PYTORCH CODE

To give a detailed picture of how the TOM architecture in this paper was implemented, the code for the forward pass of the model implemented in pytorch (Paske et al., 2017) is given in Figure 7. For efficiency, TOM is implemented with Conv1D layers with kernel size 1 instead of Dense layers.

```python
def forward(self, input_batch, input_contexts, output_contexts):

    # Setup encoder inputs
    batch_size = input_batch.shape[0]
    x = input_batch.unsqueeze(1)
    z = input_contexts.expand(batch_size, -1, -1)

    # Apply encoder
    x = self.encoder_film_layer(x, z)
    x = self.dropout(x)
    for block in self.encoder_blocks:
        x = block(x, z)

    # Aggregate state over variables
    x = torch.sum(x, dim=-1, keepdim=True)

    # Apply model core
    for block in self.core_blocks:
        x = block(x)

    # Setup decoder inputs
    x = x.expand(-1, -1, output_contexts.shape[-1])
    z = output_contexts.expand(batch_size, -1, -1)

    # Apply decoder
    for block in self.decoder_blocks:
        x = block(x, z)
    x = self.dropout(x)
    x = self.decoder_film_layer(x, z)

    # Remove unnecessary channels dimension
    x = torch.squeeze(x, dim=1)

    return x
```

Figure 7: Pytorch code for the forward pass of the TOM implementation.

This approach enables the model to run the encoder and decoder on all variables in parallel. The fact that Conv layers are so highly optimized in pytorch makes the implementation substantially more efficient than with Dense layers. In this code, `input_batch` has shape (batch size, input variables), `input_contexts` has shape (1, VE dim, # input variables), and `output_contexts` has shape (1, VE dim, # output variables). Code for TOM will be available at `https://github.com/leaf-ai/tom-release`.

## C  ADDITIONAL EXPERIMENTAL DETAILS

A sigmoid layer is applied at the end of the decoder for the CIFAR experiments, to squash the output between 0 and 1.

For the CIFAR and Daily Temperature experiments, a subset of the variables is sampled each iteration to be used as input. This subset is sampled in the following way: (1) Sample the size $k$ of the subset uniformly from $[1, n_t]$, where $n_t$ is the number of variables in the experiment; (2) Sample a subset of variables of size $k$ uniformly from all subsets of size $k$. This sampling method ensures that every subset size has an equal chance of getting selected, so that the universe is not biased towards tasks of a particular size. E.g., if instead the subset were created by sampling each variable independently with probability $p$, then the subset size would concentrate tightly around $pn_t$.

For classification tasks, each class defines a distinct output variable, i.e., a $K$-class classification task has $K$ output variables. The squared hinge loss was used for classification tasks (Janocha &

Czarnecki, 2017). It is preferable to categorical cross-entropy loss in this setting, because it does not require taking a softmax across output variables, so the outputs are kept separate. Also, the loss becomes exactly zero once a sample is learned strongly, so that the model does not continue to overfit as remaining samples and tasks are learned.

The number of blocks in the encoder, core, and decoder is $N = 3$ for all problems except UCI-121, for which it is $N = 10$. All experiments use a hidden size of 128 for all dense layers aside from the final decoder layer that maps to the output space.

The batch size was 32 for CIFAR and Daily Temperature, and $\max(200, \text{\# train samples})$ for all other tasks. At each step, $T_o$ tasks are uniformly sampled from the set of all tasks, and gradients are summed over a batch for each task in the sample. $T_o = 1$ in all experiments except UCI-121, for which $T_o = 32$.

To allow for multi-task training with datasets of varying numbers of samples, we say the model has completed one epoch each time it is evaluated on the validation set. An epoch is 1000 steps for CIFAR, 100 steps for Daily Temperature, 1K steps for Transposed Gaussian Process, 1K steps for Concentric Hyperspheres, and 10K steps for UCI-121.

For CIFAR, the official training and test splits are used for training and testing. No validation set is needed for CIFAR, because none of the models can overfit to the training set. For Daily Temperature, the second-to-last year of data is withheld for validation, and the final year is withheld for testing. The UCI-121 experiments use the preprocessed versions of the official train-val-test splits (https://github.com/bioinf-jku/SNNs/tree/master/UCI).

Adam is used for all experiments, with all parameters initialized to their default values. In all experiments except UCI-121, the learning rate is kept constant at 0.001 throughout training. In UCI-121, the learning rate is decreased by a factor of two when the mean validation accuracy has not increased in 20 epochs; it is decreased five times; model training stops when it would be decreased a sixth time. Models are trained for 500K steps for CIFAR, 100K steps for Daily Temperature, and 250K for Transposed Gaussian Process and Concentric Hyperspheres. The test performance for each task is its performance on the test set after the epoch of its best validation performance.

Weights are initialized using the default pytorch initialization (aside from the SkipInit $\alpha$ scalars, which are initialized to zero (De & Smith, 2020)). The experiments in Section 5.1 use no weight decay; in Section 5.2 use weight decay of $10^{-4}$; and in Section 5.3 use weight decay of $10^{-5}$. Dropout is set to 0.0 for CIFAR, Daily Temperature, and Concentric Hyperspheres; and 0.5 for Transposed Gaussian Process and UCI-121.

In UCI-121, fully-trained MTL models are finetuned to tasks with more than 5,000 samples, using the same optimizer configuration as for joint training, except the steps-per-epoch is set to $\lceil \text{\# train samples}/\text{batch size} \rceil$, the learning rate is initialized to 0.0001, the patience for early stopping is set to 100, and the validation performance is smoothed over every 10 epochs (simple moving average), following the protocol used to train single-task models in prior work (Klambauer et al., 2017).

TOM uses a VE size of $C = 2$ for all experiments, except for UCI-121, where $C = 128$ in order to accommodate the complexity of such a large and diverse set of tasks. For Figure 6c, t-SNE (van der Maaten & Hinton, 2008) was used to reduce the dimensionality to two. t-SNE was run for 10K iterations with default parameters in the scikit-learn implementation (Pedregosa et al., 2011), after first reducing the dimensionality from 128 to 32 via PCA. Independent runs of t-SNE yielded qualitatively similar results.

Autoencoding (i.e., predicting the input variables as well as unseen variables) was used for CIFAR, Daily Temperature, and Transposed Guassian Process; it was not used for Concentric Hyperspheres or UCI-121.

The Soft Layer Ordering architecture follows the original implementation (Meyerson & Miikkulainen, 2018). There are four shared ReLU layers, each of size 128, with dropout after each to ease sharing across different soft combinations of layers.

In Tables 2 and 3 means and standard error for each method are computed over ten runs.

The Daily Temperature dataset was downloaded from https://raw.githubusercontent.com/jbrownlee/Datasets/master/daily-min-temperatures.csv.

# D    ADDITIONAL DETAILED RESULTS FOR UCI-121 EXPERIMENT

Table 6 contains test accuracies for each UCI-121 task for all methods run in the experiments in Section 5.3.

Table 6: Accuracies for each UCI-121 task.

| Method | DR-STL | TOM-STL | DR-MTL | SLO | TOM |
|---|---|---|---|---|---|
| abalone | 65.421 | 64.464 | 64.847 | 66.667 | 65.230 |
| acute-inflammation | 100.000 | 46.667 | 100.000 | 100.000 | 100.000 |
| acute-nephritis | 100.000 | 96.667 | 100.000 | 100.000 | 100.000 |
| adult | 84.890 | 84.319 | 84.411 | 85.216 | 85.763 |
| annealing | 73.000 | 54.000 | 76.000 | 75.000 | 31.000 |
| arrhythmia | 68.142 | 58.407 | 69.027 | 58.407 | 65.487 |
| audiology-std | 36.000 | 0.000 | 76.000 | 80.000 | 68.000 |
| balance-scale | 89.744 | 46.154 | 91.667 | 96.154 | 94.872 |
| balloons | 75.000 | 75.000 | 50.000 | 75.000 | 100.000 |
| bank | 90.177 | 90.177 | 89.469 | 90.265 | 88.496 |
| blood | 76.471 | 75.936 | 74.332 | 77.005 | 72.727 |
| breast-cancer | 69.014 | 70.423 | 66.197 | 66.197 | 71.831 |
| breast-cancer-wisc | 97.714 | 97.143 | 97.714 | 97.714 | 97.143 |
| breast-cancer-wisc-diag | 97.183 | 98.592 | 98.592 | 98.592 | 97.887 |
| breast-cancer-wisc-prog | 75.510 | 77.551 | 81.633 | 73.469 | 73.469 |
| breast-tissue | 53.846 | 30.769 | 65.385 | 69.231 | 76.923 |
| car | 97.454 | 91.898 | 87.731 | 99.074 | 99.306 |
| cardiotocography-10clases | 81.168 | 79.661 | 76.083 | 82.863 | 84.557 |
| cardiotocography-3clases | 90.584 | 90.207 | 88.701 | 93.409 | 93.974 |
| chess-krvk | 35.358 | 71.842 | 35.729 | 68.848 | 66.382 |
| chess-krvkp | 98.874 | 99.875 | 98.373 | 99.625 | 99.625 |
| congressional-voting | 55.046 | 61.468 | 62.385 | 61.468 | 61.468 |
| conn-bench-sonar-mines-rocks | 86.538 | 78.846 | 78.846 | 80.769 | 84.615 |
| conn-bench-vowel-deterding | 65.801 | 12.338 | 74.675 | 98.701 | 98.485 |
| connect-4 | 79.271 | 91.456 | 76.790 | 87.122 | 87.157 |
| contrac | 57.880 | 53.804 | 55.707 | 58.967 | 55.435 |
| credit-approval | 80.814 | 81.977 | 82.558 | 83.721 | 81.977 |
| cylinder-bands | 69.531 | 63.281 | 72.656 | 72.656 | 75.781 |
| dermatology | 93.407 | 39.560 | 97.802 | 96.703 | 97.802 |
| echocardiogram | 75.758 | 69.697 | 87.879 | 75.758 | 87.879 |
| ecoli | 88.095 | 48.810 | 85.714 | 82.143 | 86.905 |
| energy-y1 | 80.208 | 80.729 | 83.854 | 95.833 | 96.354 |
| energy-y2 | 82.292 | 81.250 | 85.938 | 90.625 | 92.188 |
| fertility | 92.000 | 88.000 | 84.000 | 88.000 | 88.000 |
| flags | 47.917 | 33.333 | 41.667 | 47.917 | 41.667 |
| glass | 50.943 | 37.736 | 66.038 | 67.925 | 66.038 |
| haberman-survival | 72.368 | 73.684 | 73.684 | 73.684 | 72.368 |
| hayes-roth | 25.000 | 60.714 | 50.000 | 53.571 | 85.714 |
| heart-cleveland | 60.526 | 56.579 | 60.526 | 61.842 | 63.158 |
| heart-hungarian | 78.082 | 79.452 | 80.822 | 78.082 | 80.822 |
| heart-switzerland | 32.258 | 54.839 | 35.484 | 45.161 | 58.065 |
| heart-va | 26.000 | 20.000 | 32.000 | 30.000 | 36.000 |
| hepatitis | 69.231 | 79.487 | 71.795 | 84.615 | 84.615 |
| hill-valley | 50.000 | 50.165 | 67.492 | 63.861 | 58.251 |
| horse-colic | 86.765 | 69.118 | 83.824 | 80.882 | 79.412 |
| ilpd-indian-liver | 71.918 | 71.918 | 60.959 | 74.658 | 66.438 |
| image-segmentation | 58.524 | 14.238 | 88.952 | 89.333 | 92.381 |
| ionosphere | 86.364 | 90.909 | 90.909 | 93.182 | 88.636 |
| iris | 89.189 | 62.162 | 97.297 | 97.297 | 97.297 |

*Continued on next page.*

| Method | DR-STL | TOM-STL | DR-MTL | SLO | TOM |
|---|---|---|---|---|---|
| led-display | 75.600 | 27.200 | 79.600 | 73.600 | 74.000 |
| lenses | 83.333 | 66.667 | 50.000 | 50.000 | 50.000 |
| letter | 95.980 | 97.480 | 87.220 | 94.580 | 94.780 |
| libras | 43.333 | 11.111 | 78.889 | 76.667 | 80.000 |
| low-res-spect | 81.955 | 56.391 | 83.459 | 82.707 | 90.977 |
| lung-cancer | 50.000 | 25.000 | 62.500 | 50.000 | 62.500 |
| lymphography | 86.486 | 56.757 | 94.595 | 86.486 | 86.486 |
| magic | 86.982 | 86.898 | 81.325 | 86.877 | 87.024 |
| mammographic | 81.250 | 82.500 | 80.833 | 82.083 | 83.750 |
| miniboone | 92.782 | 94.630 | 93.345 | 94.338 | 93.532 |
| molec-biol-promoter | 88.462 | 50.000 | 69.231 | 61.538 | 92.308 |
| molec-biol-splice | 85.696 | 92.723 | 86.324 | 85.822 | 93.350 |
| monks-1 | 65.509 | 50.000 | 71.991 | 86.574 | 80.787 |
| monks-2 | 40.509 | 67.130 | 62.731 | 64.583 | 62.500 |
| monks-3 | 74.306 | 52.778 | 66.898 | 68.981 | 58.102 |
| mushroom | 99.655 | 100.000 | 99.803 | 100.000 | 100.000 |
| musk-1 | 83.193 | 57.143 | 92.437 | 90.756 | 91.597 |
| musk-2 | 98.666 | 98.848 | 98.787 | 99.272 | 99.636 |
| nursery | 99.568 | 99.877 | 95.926 | 99.753 | 99.630 |
| oocytes_merluccius_nucleus_4d | 83.922 | 70.588 | 77.647 | 83.529 | 85.098 |
| oocytes_merluccius_states_2f | 89.412 | 92.549 | 94.510 | 92.157 | 95.294 |
| oocytes_trisopterus_nucleus_2f | 73.684 | 75.877 | 75.439 | 78.509 | 78.947 |
| oocytes_trisopterus_states_5b | 94.298 | 92.544 | 93.421 | 94.737 | 92.982 |
| optical | 95.993 | 95.326 | 94.658 | 94.380 | 95.938 |
| ozone | 97.161 | 97.161 | 97.161 | 97.161 | 97.161 |
| page-blocks | 95.468 | 96.199 | 94.371 | 96.272 | 96.345 |
| parkinsons | 89.796 | 75.510 | 83.673 | 87.755 | 83.673 |
| pendigits | 96.855 | 97.055 | 97.055 | 96.884 | 96.627 |
| pima | 71.875 | 71.875 | 73.438 | 75.521 | 76.562 |
| pittsburg-bridges-MATERIAL | 73.077 | 76.923 | 88.462 | 84.615 | 92.308 |
| pittsburg-bridges-REL-L | 69.231 | 65.385 | 65.385 | 73.077 | 61.538 |
| pittsburg-bridges-SPAN | 52.174 | 56.522 | 65.217 | 65.217 | 60.870 |
| pittsburg-bridges-T-OR-D | 84.000 | 88.000 | 84.000 | 84.000 | 88.000 |
| pittsburg-bridges-TYPE | 38.462 | 50.000 | 61.538 | 65.385 | 53.846 |
| planning | 64.444 | 71.111 | 71.111 | 68.889 | 71.111 |
| plant-margin | 76.750 | 6.750 | 71.250 | 69.500 | 74.000 |
| plant-shape | 39.000 | 20.750 | 31.500 | 65.750 | 70.500 |
| plant-texture | 74.250 | 4.000 | 69.750 | 69.000 | 77.250 |
| post-operative | 72.727 | 72.727 | 77.273 | 72.727 | 72.727 |
| primary-tumor | 45.122 | 30.488 | 47.561 | 47.561 | 51.220 |
| ringnorm | 95.027 | 98.108 | 84.324 | 96.054 | 98.324 |
| seeds | 80.769 | 80.769 | 86.538 | 94.231 | 92.308 |
| semeion | 95.729 | 92.462 | 94.724 | 88.693 | 94.472 |
| soybean | 65.426 | 18.617 | 89.628 | 82.979 | 83.777 |
| spambase | 93.826 | 92.609 | 92.609 | 93.478 | 93.913 |
| spect | 61.828 | 56.989 | 67.204 | 65.054 | 68.280 |
| spectf | 49.733 | 91.979 | 60.963 | 60.428 | 91.979 |
| statlog-australian-credit | 66.860 | 68.023 | 68.023 | 63.372 | 62.209 |
| statlog-german-credit | 73.600 | 76.000 | 74.400 | 76.800 | 74.800 |
| statlog-heart | 89.552 | 79.104 | 89.552 | 82.090 | 83.582 |
| statlog-image | 96.360 | 95.841 | 90.988 | 97.054 | 97.747 |
| statlog-landsat | 89.900 | 91.250 | 83.450 | 88.950 | 90.600 |
| statlog-shuttle | 98.621 | 99.945 | 98.021 | 99.910 | 99.945 |
| statlog-vehicle | 73.934 | 48.341 | 78.199 | 79.621 | 74.882 |
| steel-plates | 74.845 | 64.536 | 68.041 | 76.495 | 77.526 |
| synthetic-control | 73.333 | 69.333 | 97.333 | 96.667 | 99.333 |

*Continued on next page.*

| Method | DR-STL | TOM-STL | DR-MTL | SLO | TOM |
|---|---|---|---|---|---|
| teaching | 60.526 | 36.842 | 55.263 | 52.632 | 47.368 |
| thyroid | 98.308 | 98.775 | 96.820 | 97.841 | 98.804 |
| tic-tac-toe | 97.071 | 97.071 | 97.071 | 97.071 | 96.653 |
| titanic | 77.636 | 77.091 | 78.364 | 78.364 | 78.364 |
| trains | 100.000 | 50.000 | 100.000 | 100.000 | 100.000 |
| twonorm | 98.270 | 98.108 | 98.162 | 98.108 | 98.054 |
| vertebral-column-2clases | 83.117 | 67.532 | 87.013 | 87.013 | 85.714 |
| vertebral-column-3clases | 70.130 | 59.740 | 84.416 | 68.831 | 85.714 |
| wall-following | 86.437 | 98.827 | 72.507 | 90.396 | 97.434 |
| waveform | 87.520 | 87.360 | 87.760 | 86.800 | 87.760 |
| waveform-noise | 85.920 | 85.360 | 85.360 | 84.720 | 85.840 |
| wine | 100.000 | 70.455 | 100.000 | 100.000 | 100.000 |
| wine-quality-red | 59.000 | 57.500 | 57.750 | 63.750 | 61.000 |
| wine-quality-white | 56.863 | 53.758 | 53.513 | 57.761 | 56.944 |
| yeast | 60.108 | 53.908 | 60.377 | 59.838 | 59.838 |
| zoo | 96.000 | 48.000 | 96.000 | 96.000 | 92.000 |

*Continued from previous page.*

