# OpenReview forum: "The Traveling Observer Model: Multi-task Learning Through Spatial Variable Embeddings"
_ICLR.cc/2021/Conference — ICLR 2021 Spotlight_

### Official Review · AnonReviewer1 · 2020-10-28
**A novel and promising framework for heterogenous multi-task learning**

**Rating:** 9
**Confidence:** 4

**Review:**

This paper presents the traveling observer model (TOM), a general framework to learn multiple heterogenous supervized (input,output) tasks, which are indexed by a continuous "variable embedding" that is automatically learned by the system. The authors show on simple problems that the learned task embeddings can recover an intuitive organization of the problems' variables in space or time. They also show that the model simultaneously trained on 121 seemingly unrelated classification tasks can outperform state-of-the art supervized methods fine-tuned on single tasks.

The proposed model is novel, technically sound, of broad interest and very promising. The paper is clearly written and easy to follow.  The presented experiments convincingly demonstrate the sensibility and usefulness of the approach. The topic perfectly fits the scope of ICLR.

Minor suggestions for improvement:
- Section 2 (first paragraph) the notations are a bit confusing here. First, the sample indices s=1...S_t are denoted as superscripts while task indices t=1...T are denoted as subscript, but then the sample indices are dropped and never used again, while task indices become superscripts and variable dimensions are denoted as subscript. The definitions of sets V_t^In and V_t^out is also strange. I think they should denote the union of all the spaces that variables are living in, but instead they are defined as finite sets of specific variables. The definition of "the universe" V in section 3 is also a bit sketchy. Is that a set of sets? a category?
- I think that when using a pre-defined "oracle" variable embedding, the proposed model becomes very similar or even equivalent to conditional neural processes (Gamello et al. 2018). It would be interesting to comment on that.
- There is an unfortunate double use of the letter h for two different things in equation (3) and (4)
- Sec. 4.4 "after joint training the model is finetuned on each task with at least 5K samples" -> is the whole model fine-tuned or only the function g? or g and h? Please clarify.

---

> ### Author Response · Authors · 2020-11-23
> **Response to AnonReviewer1**
>
> Thanks for the feedback and the praise! We are especially happy to hear that you think the paper is of broad interest and easy-to-follow. We have updated the submission based on your “Minor suggestions for improvement” and describe how each suggestion was addressed below:
>
>
> - Section 2 (first paragraph) the notations are a bit confusing here. First, the sample indices s=1...S_t are denoted as superscripts while task indices t=1...T are denoted as subscript, but then the sample indices are dropped and never used again, while task indices become superscripts and variable dimensions are denoted as subscript. The definitions of sets V_t^In and V_t^out is also strange. I think they should denote the union of all the spaces that variables are living in, but instead they are defined as finite sets of specific variables. The definition of "the universe" V in section 3 is also a bit sketchy. Is that a set of sets? a category?
>
> Based on these comments, we have updated the notation to be consistent and clear throughout the paper, including standardizing the subscripts/superscripts, removing the V_t^In and V_t^out notation, and clarifying the definition of V as a set in Section 3.
>
>
> - I think that when using a pre-defined "oracle" variable embedding, the proposed model becomes very similar or even equivalent to conditional neural processes (Gamello et al. 2018). It would be interesting to comment on that.
>
>
> This is a very interesting insight. Although CNPs were developed for a different training setting and to address a set of issues distinct from those motivating TOM, the functional decomposition of their architectures is analogous at a high level. That is, replacing the VEs in Eq. 2 with input samples and the variables with output samples yields a function that generates a prediction model given a dataset. This analogy suggests that benefits from CNPs could be adapted to TOM, such as rich uncertainty information in predictions. We think this is a very promising area of future work and have added this discussion to the third paragraph of our Discussion and Future Work section.
>
>
> - There is an unfortunate double use of the letter h for two different things in equation (3) and (4)
>
> Good catch; the g and h in Eq. 4 are now changed to g_1 and g_2, which emphasizes the fact that they are a decomposition of the original g in Eq. 2.
>
>
> - Sec. 4.4 "after joint training the model is finetuned on each task with at least 5K samples" -> is the whole model fine-tuned or only the function g? or g and h? Please clarify.
>
> We have added clarification in Sec. 4.4 that the whole model is fine-tuned.
>
>
> Again, thanks for the feedback. We are glad that you enjoyed the paper.

---

### Official Review · AnonReviewer2 · 2020-11-03
**Paper is very well written and addresses an important topic; using Traveling Observer Model in multi-task learning for tasks that do not have no spatial organization unlike, for example, images.**

**Rating:** 9
**Confidence:** 4

**Review:**

Paper is very well written and addresses an important topic; using Traveling Observer Model (TOM) in multi-task learning for tasks that do not have no spatial organization unlike, for example, images. Although the paper is said to be a first implementation of TOM, it does thorough experimenting and result analysis of its preformance from various aspects and by comparing it to many sophisticated models. Future research for improving and testing the algorithm is clearly detailed.

Related scientific literature is sufficiently addressed,  mathematical background and the method are clearly presented, extensive and relevant experiments are done and result analyzed.

 I didn't even find any typos.

---

> ### Author Response · Authors · 2020-11-23
> **Response to AnonReviewer2**
>
> Thanks for the feedback! We are glad that you agree that this is an important topic and that you found the paper clear and comprehensive.

---

### Official Review · AnonReviewer5 · 2020-11-05
**Confused in the experimental setting.**

**Rating:** 6
**Confidence:** 4

**Review:**

This paper tries to solve a multitask learning task by building a task embedding use the training data and use a shared decoder to predict new data.

The paper is well structured and easy to understand the general idea. This idea that maps the observed input and output into a shared space as the task embedding which is good. However, I start to get confused when I start to fetch more details and intuition behind this. I list some of my puzzles below and hope to hear from the authors:

1. The tasks could be unrelated. Do they need to have the same output dimension? Otherwise, you cannot use the same decoder for all the tasks.
2. When you train the g, are you still using observable data?
3. I am a little puzzled about the meaning of x,  y,  z in each task. In the toy example (Figure 1). Are all x, y, z the positions? Do we need to predict the value given z or to predict y give z? What is the value mean in Figure 1? Meanwhile, can you explicitly introduce what x, y, z represents in other experiments?
4. What does this sentence mean "the z’s are not known a priori, but they can be learned alongside f and g by gradient descent"(10th line on page two), from my perspective, z should be the input or the transformation of input, right? Because in equation 4, x_j is shown on the left side but the right side only contains z_j.

The name of the model is somehow misleading: TRAVELING OBSERVER MODEL. Travel has the meaning of time flow. But it seems that you can see all the training data (observation) at any time (e.g. the last experiment).

It will be better if the authors explain each abbreviation before using them, for example
- VEs:  Does it mean variable embedding?
- HW, LN, MS, SNN in Table 4.

Can the proposed method compare with some meta-learning baseline? For example[1], this paper is targeted at multimodal learning. They also have task embedding for each task and borrow the idea from FILM. Their task embedding also depends on the observable data.  They use meta loss to train the model.

I am sorry if I missed any part which has been explained in the paper. Looking forward to your reply.


[1]Multimodal Model-Agnostic Meta-Learning via Task-Aware Modulation

---

> ### Author Response · Authors · 2020-11-23
> **Response to AnonReviewer5 (1/2)**
>
> Thanks for the feedback! We have updated the submission accordingly. The most significant changes were that we added a figure at the beginning of the paper and revised the notation to make the setup clear. Each specific comment is addressed below:
>
>
> “1. The tasks could be unrelated. Do they need to have the same output dimension? Otherwise, you cannot use the same decoder for all the tasks.”
>
> The tasks do not need to have the same output dimension. For any given output dimension, i.e., number of output variables, the decoder g is applied once for each output variable. We have added a figure to the beginning of the paper to make this setup clear, and we have moved a sentence from the Appendix to the first paragraph of Section 5.2 noting how each class defines an output variable in the case of classification tasks.
>
>
> “2. When you train the g, are you still using observable data?”
>
> Yes, all parameters of the model are trained at the same time. That is, f, g, and all the variable embeddings (i.e., the z’s) are all trained jointly with gradient descent in all the experiments. We have updated the notation to make this setup clear.
>
>
> “3. I am a little puzzled about the meaning of x, y, z in each task. In the toy example (Figure 1). Are all x, y, z the positions? Do we need to predict the value given z or to predict y give z? What is the value mean in Figure 1? Meanwhile, can you explicitly introduce what x, y, z represents in other experiments?”
>
> The new figure in the beginning of the paper should make this setup clear. We have also revised the notation to be simpler and consistent throughout the paper, which should make it easier to understand the meaning of the x’s, y’s, and z’s.
>
> In short, in all tasks, the x_i are simply the input variables of the task, and the y_j are the output variables of the task. These are the usual input and output variables in supervised learning. In all experiments, the z’s are the variable embeddings, and there is one associated with each input variable and one associated with each output variable. For example, in the toy Example (now Figure 5), the z’s are the locations of the x’s and y’s, and Value is the scalar data for the x’s and y’s in the dataset.
>
>
> “4. What does this sentence mean "the z’s are not known a priori, but they can be learned alongside f and g by gradient descent"(10th line on page two), from my perspective, z should be the input or the transformation of input, right? Because in equation 4, x_j is shown on the left side but the right side only contains z_j.”
>
> The z’s are learned embedding vectors, like word embeddings learned by word2vec and other NLP models, and like the learned task embeddings in methods discussed in the third paragraph of Section 2. Unlike the task embeddings used in the paper you’ve referenced [1], the task embeddings of Table 1b are not generated in a forward pass through the dataset, but are trained directly for each task by gradient descent. So yes, the z’s are inputs to the model, but unlike the input x’s that comes directly from the dataset, the z’s are randomly initialized and learned with the rest of the model parameters. We believe that our updated notation as well as the new Figure 1 and its explanation makes the relationship between the x’s, y’s, and z’s clear.
>
>
> “The name of the model is somehow misleading: TRAVELING OBSERVER MODEL. Travel has the meaning of time flow. But it seems that you can see all the training data (observation) at any time (e.g. the last experiment).”
>
> Yes, the word Traveling is used here abstractly, to convey the idea that observations are made at different locations in the embedding space before a prediction can be made. The observations could exist at different temporal locations (as in the Daily Temperature problem), but in general they are simply at different locations in the embedding space. We believe the new figure at the beginning of the paper makes this clear.
>
>
> “It will be better if the authors explain each abbreviation before using them, for example
> 	•	VEs: Does it mean variable embedding?
> 	•	HW, LN, MS, SNN in Table 4.”
>
> Yes, “VEs” stands for variable embeddings. Based on your feedback, in the updated version of the paper we introduce the abbreviation “VEs” the first time “variable embeddings” is used, i.e., in Section 1.
>
> The caption of Table 4 now points to a reference where the details of HW, LN, MS, and SNN are described. A full description of these methods is outside the scope of this paper; they are only related to TOM in that they are deep learning approaches that have been applied to this set of tasks. However, if you think it would be helpful, we would be happy to include a description of these methods in the Appendix.

---

> > ### Author Response · Authors · 2020-11-23
> > **Response to AnonReviewer5 (2/2)**
> >
> > “Can the proposed method compare with some meta-learning baseline? For example[1], this paper is targeted at multimodal learning. [1]Multimodal Model-Agnostic Meta-Learning via Task-Aware Modulation”
> >
> > Thanks for the interesting reference. As mentioned in the Discussion and Future Work section of the paper, we agree that extending TOM to the meta-learning setting is a promising area of future work. The reference you mention does not handle the disjoint input/output issue that is the main motivation for TOM. However, there are still some interesting connections, which we specify in two places in the paper: (1) in the Instantiation section, as an additional example of a model that uses FiLM to incorporate task embeddings; and (2) in the third paragraph of Discussion and Future Work, where we discuss an intriguing analogy to models like Conditional Neural Processes that was brought up by AnonReviewer1.
> >
> > “I am sorry if I missed any part which has been explained in the paper. Looking forward to your reply.”
> >
> > Thanks for the questions and suggestions!

---

### Official Review · AnonReviewer4 · 2020-11-06
**Embedding for multi-task learning problems with disjoint inputs**

**Rating:** 6
**Confidence:** 3

**Review:**

**Summary.** Authors present a methodology for performing multi-task learning from data with disjoint and heterogeneous input domains. Particularly, they introduce an embedding of the inputs, in order to project each pair of input-output observations in a common continuous manifold where the exploration is significantly easier. Results show that the approach is valid with both synthetic and real-world data and they also demonstrate that the model is flexible when increasing/decreasing the dimensionality of the latent manifold.

**Strengths.** The explanation of the multi-task learning scenario with disjoint input domains is particularly well-written. This description makes easier to understand the reasons behind the introduction of the embedding between every single input and latent vectors z. Additionally, authors did an effort for explaining point-by-point the structure of the deep NN transformation behind the embedding. This is valuable. I appreciated the design of experiments and (author-blind) video on youtube was impressive.

**Weaknesses, Questions & Recommendations.**
The main weaknesses (to me) in the paper are:
[W1]. There is likely a lack of references and analysis about similar works on multi-task learning with the particular problem of disjoint inputs. This makes the reader doubt about the potential novelty of the model, in particular about the embedding.
[W2]. The notation based on subsets V_t is a bit confusing, (I think that keeping the (x,y,z) notation all along the paper would be better). Particularly in the pp.3, this notation is a difficult to follow before the introduction of the TOM embedding.
[W3]. The TOM implementation may be better placed before the experiments, being a bit better connected with the main section of the manuscript, but this is just an opinion.
[W4]. More analysis on the dimensionality D of the manifold could be of interest for the reader. In the last experiment, this dimensionality is pretty high. [Q] Why is this? What is the principal consequence?
[W5]. Error metrics in the experiments do not include confidence intervals or variance values from several runs.
[W6]. Typically, one chooses Discussion or Conclusion. The content of the Conclusion is similar to the thing said in the previous section.

Recommendations:
[Rec1]. Motivating even better the disjoint input problem from the very beginning would make the paper stronger.
[Rec2]. An input-output notation all along the paper and some diagram explaining the projection into a continuous manifold would help as well.
[Rec3]. Details about the implementation could be better placed in the appendix, or at least integrated with the model and the flow of explanations.
[Rec4]. Confidence intervals in the tables of error metrics as well as a bit more of motivation for the circle experiment would improve the presentation of experiments.

**Reasons for score.** I understood the idea that authors presented and the problem of disjoint input domains. However, I feel that the presentation of the model is a bit weak as well as the experiments could be improved with a few details. The last pp. of the manuscript with the duplicity Discussion+Conclusion is also a bit odd. For this reason, I cannot recommend an acceptance score for this venue.

**Post-rebuttal comments.** Thanks to the authors for their response. The updated version of the manuscript addressed my main concerns and recommendations. Now, it is clearly improved, figures and metrics updated and the proposed methodology is better presented. Authors even did major changes on the structure of the paper, what I recognize as an important revision. Having said this, I raised my score.

---

> ### Author Response · Authors · 2020-11-23
> **Response to AnonReviewer4**
>
> Thanks for the feedback! We have updated the submission based on your comments. We discuss how each comment was addressed below:
>
>
> [W1]. “There is likely a lack of references and analysis about similar works on multi-task learning with the particular problem of disjoint inputs.”
>
> The paper includes all references we are aware of on Deep MTL with fundamentally disjoint inputs; the area is highly unexplored. Based on your suggestion, we have expanded the discussion of such existing methods in the fourth paragraph of Section 2 (characterized by Table 1c) to give a more complete picture of the existing methods.
>
>
> [W2-Rec2]. “The notation based on subsets V_t is a bit confusing, (I think that keeping the (x,y,z) notation all along the paper would be better).” “An input-output notation all along the paper and some diagram explaining the projection into a continuous manifold would help as well.”
>
> We have made the notation consistent along the lines of what you suggested, and we agree it improves the readability and cohesion of the paper. We have also added a new figure 1 at the beginning of the paper that clearly shows how TOM makes observations and predictions via the manifold.
>
>
> [W3-Rec3]. “The TOM implementation may be better placed before the experiments, being a bit better connected with the main section of the manuscript, but this is just an opinion.”
>
> This is a nice idea; we have moved the implementation to its own section as suggested.
>
>
> [W4]. “More analysis on the dimensionality D of the manifold could be of interest for the reader. In the last experiment, this dimensionality is pretty high. [Q] Why is this? What is the principal consequence?”
>
> We agree this is a very natural question. As noted in Section 5.3, the dimensionality is higher in this experiment because it is a real-world scale-up from the earlier experiments: the datasets are so diverse that larger embeddings are required to handle this diversity. The dimensionality was set to 128 to match the number of task-specific parameters of the other Deep MTL comparison methods. The goal of the paper is to establish the foundations of TOM, and show that it can work, and we expect future work to explore the various dimensions of the model more fully, such as the VE dimensionality as you suggest. To that end, we ran preliminary experiments with varying dimensionality, and the results indeed suggest that the dimensionality could be reduced while maintaining high performance. These results are now included in Appendix A.
>
>
> [W5-Rec4]. “Confidence intervals in the tables of error metrics as well as a bit more of motivation for the circle experiment would improve the presentation of experiments.”
>
> We completed additional runs and added confidence metrics to the tables. We also added additional motivation to both the Daily Temperature and Concentric Hyperspheres experiments.
>
>
> [W6]. “The content of the Conclusion is similar to the thing said in the previous section.”
>
> Thanks for pointing on the redundant phrasing; We have updated the Discussion section to minimize the redundancy.
>
>
> [Rec1]. “Motivating even better the disjoint input problem from the very beginning would make the paper stronger.”
>
> The new figure at the beginning of the paper more clearly motivates the disjoint tasks setting, and the additional details added to Section 2 on the disjoint task setting also help motivate the problem in the context of existing work.
>
>
> Again, thanks for the feedback. We believe the paper is now more clear as a result.

---

### Decision · Program_Chairs · 2021-01-07
**Final Decision**

**Decision:**

Accept (Spotlight)

**Comment:**

Post rebuttal, the reviewers all recommend acceptance.